# Caffeine Improves Sprint Time in Simulated Freestyle Swimming Competition but Not the Vertical Jump in Female Swimmers

**DOI:** 10.3390/nu16091253

**Published:** 2024-04-23

**Authors:** Kürşat Acar, Ahmet Mor, Hakkı Mor, Zehra Kargın, Dan Iulian Alexe, Mekki Abdioğlu, Raci Karayiğit, Cristina Ioana Alexe, Adin Marian Cojocaru, George Danuț Mocanu

**Affiliations:** 1Department of Physical Education and Sports, Faculty of Sports Sciences, Sinop University, 57010 Sinop, Turkey; kursatacar@sinop.edu.tr (K.A.); amor@sinop.edu.tr (A.M.); zkargin@sinop.edu.tr (Z.K.); 2Department of Coaching Education, Faculty of Sport Sciences, Ondokuz Mayıs University, 55280 Samsun, Turkey; hakki.mor@omu.edu.tr; 3Department of Physical and Occupational Therapy, “Vasile Alecsandri” University of Bacau, 600115 Bacau, Romania; 4Department of Sports Sciences, Institute of Health Sciences, Ankara University, 06110 Ankara, Turkey; 5Faculty of Sport Sciences, Ankara University, 06830 Ankara, Turkey; rkarayigit@ankara.edu.tr; 6Department of Physical Education and Sports Performance, “Vasile Alecsandri” University of Bacau, 600115 Bacau, Romania; alexe.cristina@ub.ro; 7Faculty of Physical Education and Sport, Spiru Haret University, 041905 Bucharest, Romania; ushefs_cojocaru.adin@spiruharet.ro; 8Individual Sports and Physical Therapy Department, “Dunărea de Jos” University of Galati, 800008 Galați, Romania; george.mocanu@ugal.ro

**Keywords:** sports nutrition, caffeine, ergogenic aid, swimming performance, jump performance

## Abstract

Caffeine (CAF) has been shown to be an effective ergogenic aid in enhancing sports performance, including vertical jump (VJ), sprint, balance, agility, and freestyle swimming performance (FSP). However, whether acute CAF supplementation improves FSP in moderately trained female swimmers has not been well documented. Therefore, this study aimed to investigate the effects of CAF intake on vertical jump, balance, auditory reaction time (ART), and swimming performance in female swimmers. In a double-blind, cross-over design, eight moderately trained female swimmers (age: 21.3 ± 1.4 years, height: 161.2 ± 7.1 cm, body mass: 56.3 ± 6.7 kg, body mass index (BMI): 21.9 ± 1.3 kg/m^2^, and habitual CAF intake: 246.4 ± 111.4 mg/day) ingested caffeine (CAF) (6 mg/kg) or a placebo (PLA) 60 min before completing VJ, balance, ART, and 25/50 m FSP. CAF supplementation resulted in a significantly lower time both in 25m (*p* = 0.032) and 50m (*p* = 0.033) FSP. However, CAF resulted in no significant difference in VJ, ART, and RPE (*p* > 0.05). Balance test results showed a non-significant moderate main effect (d = 0.58). In conclusion, CAF seems to reduce time in short-distance swimming performances, which could be the determinant of success considering the total time of the race. Thus, we recommend coaches and practitioners incorporate CAF into swimmers’ nutrition plans before competitions, which may meet the high performance demands.

## 1. Introduction

Caffeine (CAF) is a psychoactive drug capable of enhancing psychological responses [1] as well as physical performance in a variety of exercise and athletic conditions, such as time trials [2], 5 km cycling [3], sprint [4,5], taekwondo-specific agility test [6], and resistance exercise performance [7]. CAF’s capacity to bind to adenosine A1A and A2A receptors is most likely responsible for these ergogenic effects. CAF’s binding to these receptors might boost alertness and reduce feelings of fatigue, which may lead to an enhanced exercise performance [8]. In addition, CAF may increase motor neuron activation and prevent the decrease in voluntary activation caused by exercise-induced exhaustion; these effects may potentially contribute to performance [9,10]. Additionally, CAF enhances the Na^+^-K^+^ pump activation and an enhanced high-intensity sprint performance was reported with increased blood epinephrine and norepinephrine concentration [11]. Through these mechanisms, ingesting 3–6 milligrams per kilogram of body mass (mg/kg) of CAF 60 min before exercise is well documented to be beneficial for athletic performance [12].

Investigations examining the effects of CAF on exercise performance have explored the impact of a number of parameters on the ergogenic response, including dosage, timing, genetic factors, training status, and mode of ingestion [13]. However, male individuals were predominantly employed in these investigations. Evidently, noticeable variations between the sexes regarding body size, lean body mass, and hormonal activity may influence the ergogenic benefits of the same caffeine dosages [6,14]. Since female participants comprised merely 10% [15], 13% [14], or ~20% [16] of the total number of participants in meta-analyses investigating the ergogenic impact of CAF in several athletic disciplines, it might be hard to generalize the prescription guidelines. This was corroborated by Sabblah et al. [17], who demonstrated that 5 mg/kg of CAF improved one repetition maximum strength performance in both males and females. However, only males showed a trend toward improvement in muscular endurance. In support of this, the benefits of CAF were shown to be more assertive in males, but a decaffeinated beverage caused larger impacts in women, i.e., less somnolence, more activation, and a balance of awareness and tiredness [18]. However, more recently, research has emerged that reveals no difference in the performance responses of men and women to CAF [2,3]. Clearly, further research including female participants is necessary in light of the contradictory findings.

Although the ergogenic effects of CAF have been extensively studied in team sports such as football, basketball, and handball, there are not much data on individual sports such as swimming [12]. CAF’s ergogenic ability for high-intensity performance may be beneficial for sprint swimmers. Furthermore, 25 m to 50 m freestyle events usually last between 20 and 45 s, and strategies are constantly being sought to improve anaerobic performance. In swimming competitions, particularly for shorter distances, the swimmer’s capability to execute a strong start accounts for 30% of a 50 m race [19]. Start performance in swimming is a combination of reaction time, vertical and horizontal force off the block, a low resistance entering the water and underwater gliding, and leg propulsion [20]. Above all, exerting force off the block and the leg propulsion phase have been linked to lower body strength [19,21].

Accordingly, lower body strength seems to be a key determinant of starting performance in sprinting [19], and greater lower body strength can induce enhancements in swimming block start performance [22]. Given the increased performance in the 50 m and 400 m average swimming time [21], the 20 m sprint swim time [23], and the 50 m front crawl performance [24] after several 6, 16, and 12 weeks, respectively, of plyometric training, the vertical jump height could be considered the best predictor of lower body muscular strength [25]. In swimming events, the final places are often decided by very small differences. This is shown by the fact that the difference between first and second place in the 200 m butterfly stroke finals at the 2016 Olympics was only 0.04 s (1:53.36 vs. 1:53.40 min). In a recent meta-analysis by Grgic [12], which included eight studies, CAF intake between 3 and 6 mg/kg doses was found to be effective for short-distance swimming events (Cohen’s d: −0.14; *p* = 0.03). However, only 22% of the participants were female in all studies and all included both males and females [12].

To the best of the authors’ knowledge, no single study has examined CAF’s benefits on female swimmers. Moreover, the lower body anaerobic performance of swimmers may have an impact on start time and overall swimming time [26]. Based on this knowledge, the current study, for the first time, was designed to investigate the effects of 6 mg/kg of CAF on 25 m and 50 m swimming sprint time, auditory reaction time, balance, and vertical jump performance in female swimmers. It was hypothesized that 6 mg/kg of CAF would enhance the lower body anaerobic and swimming sprint performances.

## 2. Materials and Methods

Eight healthy, non-smoking, young, moderately trained [27] female swimmers participated in this study (Table 1). The required sample size was estimated using G*Power software (Heinrich-Heine-University Düsseldorf, version 3.1.9.2, Düsseldorf, Germany). A sample size of 8 participants was determined sufficient. (Effect size: 0.50, Confidence interval: 1 − β 0.95, Error: α 0.05, and Actual power: 0.96). All participants had 8.5 ± 1.4 years of competitive swimming experience at club standard and at least 3 ± 4 years of experience swimming in regional and university-level leagues. Participants completed at least 5 weekly swimming training sessions (7.5 h a week). The inclusion criteria were as follows: (a) taken no ergogenic aids within three months before the tests, which could potentially affect the results; (b) at least 3 years of swimming training experience; (c) had no metabolic, cardiovascular, or respiratory diseases; and (d) had no orthopedic injury within the previous 6 months that could influence performance outcomes. All participants were informed of the experimental procedures before giving their written informed consent. This research was carried out in accordance with the Declaration of Helsinki and Ethical approval for this study was received from the Human Research Ethics Committee at Sinop University (Reference number: E-57452775-050.01.04-104437).

### 2.1. Experimental Design

On the first visit, a habitual caffeine consumption questionnaire was applied in addition to participants’ anthropometric and body composition assessments (e.g., body mass and stature). Habitual caffeine intake was determined through a validated questionnaire [28]. Only participants with a daily CAF intake of less than 250 mg·d^−1^ were included, in order to control individual differences in responsiveness to CAF from habituation. On this visit, participants also performed the testing protocol for familiarization at a low intensity that would not make them exert vigorous effort. Following completion of this initial familiarization, participants were assigned to ingest either CAF or PLA in a double-blind, cross-over, randomized counterbalanced design. There were at least 48 h between sessions to ensure CAF had a washout and to allow participants to recover completely. Participants were asked to record their diet 24 h before the first test session and replicate it 24 h before the second test session. For 24 h before and for each of the testing sessions, participants were asked to refrain from ingestion of CAF, ergogenic aid (e.g., nitrate and sodium bicarbonate), alcohol, and anti-inflammatory drugs; not to engage in strenuous physical activity; and to be strict with their nutrition and rest.

Regarding the circadian rhythm, the tests and measurements were applied to the participants at the same time of the day (between 1 and 3 p.m.), under similar environmental conditions (pool: ambient temperature +37 °C, pool water temperature 26–27 °C, pH 7.5, chlorine 1.5 laboratory: ambient temperature 22 ± 0.2 °C, humidity 63 ± 0.4%, pressure 1017 ± 0.7 mbar; and mean ± SD) in the university’s performance laboratory and semi-Olympic indoor swimming pool. Participants were instructed to wear the same clothing and footwear to all the testing sessions. Participants completed a 15 min standardized warm-up. Ad libitum water consumption was allowed in both trials. The testing protocol in each experimental session consisted of VJ, balance, ART, and 25/50 m FSP, respectively. These performance tests were employed because they were the same as swimmers’ moves in training and competitions. A 3 min passive rest period was given between the performance tests (except the vertical jump test) to facilitate recovery. In addition, the Borg scale ranging from 0 to 10 [29] was used to measure the rate of perceived exertion (RPE) at the end of the tests. A Schematic diagram of the experimental protocol is displayed in Figure 1.

### 2.2. Procedures

#### 2.2.1. Anthropometric and Body Composition Assessments

All body composition and anthropometric measurements were conducted during the initial familiarization session. Body mass was obtained in kg with a bioelectric impedance analysis device (BIA, Inbody 120, InBody Co., Ltd., Seoul, Republic of Korea), and height was obtained in cm with a portable stadiometer (Seca 213, Hamburg, Germany). The body mass indexes of the athletes were determined as follows: after taking the height and body weight values, they were calculated by dividing the body weight by the square of the height in meters (kg/m^2^).

#### 2.2.2. Balance Test

A portable dynamic balance device (Togu Challenge Disc 2.0, Prien am Chiemsee, Rosenheim, Germany) was utilized to assess the balance of the participants. The platform was free to move in all directions (up to a maximum of 12°) and thus provided an unstable ground. The challenge disc recorded the athlete’s movements with three-dimensional motion sensors and sent the data in real time to its software on the smartphone or tablet via Bluetooth. Stability index ranges were categorized into 1 to 5 (1—very good, 2—good, 3—normal, 4—weak, and 5—very weak), and a lower score (*p*) indicated a better balance. Initially, the researcher showed the application on the tablet at eye level to the athlete, and the athlete stood barefoot on the platform to eliminate the possible effects of different types of shoes on the results. Later, the athletes were instructed to stand in the middle of the disc and keep their balance for 20 s (after 10 s of preparation, 5 s of which is a countdown) with their arms free to swing. During the test, participants were told to keep the point in the circle as central and stable as possible. The platform provided a safe measurement for athletes with its non-slip surface. The test was performed two times with a 3 min passive rest, and the best score was used as the dynamic balance test score [30].

#### 2.2.3. Auditory Reaction Time Test

The ART of the participants was measured using a specialized software program (https://cognitivefun.net, accessed on 3 June 2022). The test was applied after providing appropriate environmental conditions for minimizing external stimuli. Prior to the test, the researcher made the application on the computer ready for measurement at a distance where the athlete could see the screen clearly. Initially, the participants were introduced to the test, and then, when they felt ready, the athletes started the test without any commands. The test was repeated twice with a 1 min passive rest period, and the best time was recorded as ART.

#### 2.2.4. Vertical Jump and Anaerobic Power Test

A digital vertical jump device (Takei 5406 Jump-MD Vertical Jumpmeter, Tokyo, Japan) was used to measure the VJ scores of the participants. Firstly, the rubber vertical jump plate was placed on a flat surface. In order to eliminate the possible effects of different types of shoes on the results, the participants were instructed to take off their shoes and stand “ready” with bare feet centered on the plate (10–20 cm from each other). Afterwards, the researcher (the same person fastened the digital belt in all trials for test reliability) zeroed the digital belt, wound it tightly around the waist of the participants, and turned the pulley gently in the direction of the arrow to take the slack out of the rope. Once the athletes were ready, they quickly moved from the upright standing position to a position of 90° flexion of the knees and free swing of the arms and jumped for the maximum height. The test was repeated if participants jumped by stepping forward, the measuring tape was loose, or they did not land on the rubber plate after jumping. Each player performed 2 trials interspersed with 1 min rest between each jump, and the best (highest) jump was recorded in cm with an accuracy of ±1 [31]. Participants’ anaerobic power calculations were executed using the Lewis formula: Anaerobic Power (W) = {√4.9 [Body Weight (kg)] √Vertical Jump (m)} [32]. 

#### 2.2.5. Swimming Performance Test

The swimming test was performed in the Sinop semi-Olympic indoor swimming pool. The water temperature of the swimming pool was held constant at 26–27 °C. For uniformity, freestyle was used by all subjects. The speed at 25 m and 50 m of maximum effort was recorded using a manual ±0.01 s precision chronometer (Casio, Tokyo, Japan). A start whistle blow signaled the start of the swimming test with the participants leaving the start block, and the finish line is indicated by reaching the opposite end of the pool for 25 m and for 50 m, it is the starting point.

### 2.3. Supplementation Protocol

Participants consumed 6 mg/kg of CAF (Nature’s Supreme, Istanbul, Türkiye) or PLA supplements (wheat bran) in capsules (8 × 1 g capsules in the same color and form) 60 min before the testing protocol [8]. A researcher, who had no further involvement in this research, prepared the caffeine dose using electronic laboratory scales with one milligram of sensitivity at room temperature.

### 2.4. Statistical Analyses

Data were checked for normality by using the Shapiro–Wilk test. Comparison between groups was analyzed with the paired sample *t*-test, which was applied to test for differences between the caffeine and placebo supplement in vertical jump and anaerobic power, balance, auditory reaction time (ART), RPE, and freestyle swimming performance tests. Cohen’s *d* was utilized in the calculation of effect size (large *d* > 0.8, moderate *d* = 0.8 to 0.5, small *d* = 0.5 to 0.2, and trivial *d* < 0.2) [33]. Statistical significance was accepted as *p* < 0.05 and all data were analyzed using SPSS 27.0 (IBM Corp., Armonk, NY, USA), and are presented as mean ± SD.

## 3. Results

No difference was observed in VJ height (42.6 ± 5.8 vs. 41.8 ± 4.1, *p* = 0.618, and d = 0.14, Table 2) and anaerobic power (*p* = 0.695, d = 0.05, Table 2) in CAF and PLA conditions. There were no significant differences (*p* > 0.05) between CAF and PLA conditions for RPE (3.0 ± 0.7 vs. 3.0 ± 0.9, *p* = 1.000, d = 0.00, Table 2) and balance (2.2 ± 0.6 vs. 1.8 ± 0.4, *p* = 0.162, d = 0.58, Table 2). Also, there were no differences between CAF and PLA conditions for ART fastest (244.2 ± 30.8 vs. 217.6 ± 78.6, *p* = 0.387, d = 0.44, Table 2), ART slowest (552.0 ± 155.3 vs. 544.1 ± 135.5, *p* = 0.926, d = 0.05, Table 2), ART average (327.6 ± 50.0 vs. 320.3 ± 28.5, *p* = 0.699, d = 0.18, Table 2), and ART deviation (115.6 ± 55.8 vs. 118.2 ± 58.6, *p* = 0.941, d = 0.04, Table 2).

The paired sample *t*-test revealed a significant difference in CAF and PLA conditions. The freestyle swimming speed was significantly improved in 25 m FSP (22.8 ± 3.2 vs. 21.7 ± 2.8, *p* = 0.032, d = 0.37, Figure 2) and 50 m FSP (50.7 ± 6.4 vs. 48.4 ± 5.8, *p* = 0.033, d = 0.38, Figure 2).

## 4. Discussion

This study aimed to investigate the effect of acute CAF intake on jumping, balance, auditory reaction time, and swimming performance in female swimmers. To the best of our knowledge, the present study is the first to investigate the effect of acute CAF intake on selected performance parameters in swimming. The main finding of this study is that 6 mg/kg of CAF intake increased swimming performance. Furthermore, no difference was noted between PLA and CAF in RPE, VJ, balance, and ART. These findings are partially in line with our experimental hypothesis that a moderate dose of CAF supplementation enhances swimming performance in female swimmers.

The present study revealed that 6 mg/kg of CAF intake increased swimming performance in female swimmers, but there was no difference in RPE. Acute CAF intake has been consistently shown to enhance swimming performance [34,35,36,37]. However, CAF appeared to demonstrate a limited ergogenic effect on swimming performance [38], while a study suggested that CAF could not improve swimming performance [39]. Salgueiro et al. [34] investigated the acute effects of 6 mg/kg CAF ingestion on ten male highly trained (≥6 sessions per week) swimmers’ average swimming speed, RPE, and lactate concentration and they suggested that CAF ingestion increased swimming speed and decreased perceptual and lactacidemic responses. In another study conducted with nine highly trained (≥8 sessions per week) freestyle swimmers, researchers reported an improved repeat sprint ability following a low dose (3 mg/kg) of acute CAF supplementation applied 60 min before exercise [37]. Likewise, 3 mg/kg of CAF reduced the time in 50 simulated swimming competitions and increased peak power in a 45 s maximal ergometer test [35]. These results are in agreement with the data obtained from the present study. Although the administered doses of CAF differed from low to moderate (3–6 mg/kg) and the participants were classified as physically active to elite levels, CAF conditions, somehow, increased performance in similar test distances (45 to 200 m). Accordingly, the study by Vanata et al. [36] conducted with 30 collegiate male and female swimmers reported significantly improved performance in a 50-yard swim time trial with 3 mg·kg^−1^ of CAF ingested 30 min before exercise, which is also in line with our study. Interestingly, however, Pruscino et al. [38] indicated that 6 mg/kg of CAF intake alone had a limited ergogenic benefit to repeated 200 m swimming performance in elite male freestyle swimmers. In addition, Alkatan [39] noted that coffee with 250 mg of CAF was ineffective in increasing swimming speed by reducing the time required for the completion of 25 m of freestyle swimming. Evidently, previous studies [34,36,37,38,39] were designed by recruiting various CAF dosing, timing, mode of ingestion as well as number of participants, training status, and sex. Therefore, it is worth noticing that the positive or negative impact of CAF intake on performance could not be elucidated by these factors. In fact, except for a study [35] with light CAF consumers (less than one can of soda or energy drink per d), all these studies suggest only one feature in common, which is that the caffeine habituations of subjects were not specified. Thus, differences in responsiveness to CAF from habituation could have determined the outcomes of these studies, which may potentially account for the present study’s results. CAF has been characterized by a molecular structure similar to adenosine. After it is present in the bloodstream, CAF binds to adenosine receptors and alleviates fatigue, which, in turn, leads to an increased exercise performance. Indeed, it has been well documented that CAF exerts its effects by blocking adenosine receptors [12] and generates ergogenic effects on the CNS through antagonism of adenosine receptors [8]. Also, the performance-enhancing effects of CAF may be associated with its role in increasing neurotransmitter release and facilitating motor unit recruitment [8,40]. Alternatively, it seems likely that these effects of CAF may explain the improvement in swimming performance in the present study. However, the underlying mechanisms behind the CAF ergogenicity remain unknown considering the paucity of studies in female swimmers. Therefore, to offer conclusive evidence, it is mandatory to investigate and elucidate the changes in the mechanism of action, and further studies are needed to assess CAF’s impact on the neuromuscular responses during anaerobic tasks in female swimmers. Additionally, short-distance swimming events are more likely to demonstrate a significant ergogenic effect of CAF ingestion on swimming performance compared to moderate-to-long swimming distance events [12]. On the contrary, CAF’s ergogenicity in RPE has been mostly observed in long-distance continuous exercises [34,41,42], while studies [37,43,44,45,46] utilizing short-distance explosive exercises reported no significant difference as in the present study. Hence, the absence of self-perception differences in RPE values may strongly be associated with the high-intensity intermittent nature of exercises and events such as sprinting, maximal effort, or resistance exercise as previously speculated [47].

The present study suggests that CAF consumption did not enhance VJ performance in female swimmers. CAF supplementation at doses ranging from 2 to 6 mg/kg was concluded to be effective in upper and lower body ballistic exercise performances, which are highly likely to be related to a better performance that determines placings in swimming competitions [48], but most recent studies with CAF administration of 3 mg/kg [49] and 6 mg/kg [50] revealed no additional positive effect on lower body power and jump performance in trained females. In another study, researchers indicated 6 mg/kg of acute CAF intake did not increase anaerobic power [51]. Together, these outcomes corroborate the results obtained from the current study. However, some evidence from several studies is not congruent with the results of the present study. It has recently been reported that acute CAF intake increased jump performance and anaerobic effort [52]. Also, De Salles Painelli et al. [53] reported improved CM performance and total repetitions. Similarly, acute CAF ingestion at low to moderate doses (2–4–6 mg/kg) in recreationally active males increased VJ performance [48]. Moreover, 5 mg/kg of CAF intake enhanced jump performance in elite male and female jumpers [54]. In volleyball players, 6 mg/kg of CAF ingestion increased CMJ height significantly [55], in addition to caffeinated chewing gum with 6.4 mg/kg CAF, which was effective in improving sport-specific jump performance [56]. Also, numerous studies reported an augmented jump performance in CAF conditions compared to PLA conditions [20,57,58,59]. The mechanism by which CAF improves jump performance is not fully known. However, there is no discrepancy present causing us to believe that CAF increases muscle fiber conduction velocity, which results in greater motor unit recruitment [9]. Given that caffeine ingestion improved short-distance swimming performance in the present study, it would therefore be expected that CAF would positively affect jump performance. During a typical training year, a swimmer goes through an extensive amount of concurrent training that comprises both pool and dryland exercises. The magnitude and type of this training schedule differ depending on the periodization of the training plan [60]. Mostly, swimmers perform nine to ten 1.5–2 h pool sessions per week and only three 1–1.5 h dryland sessions [61]. Consequently, swimmers become familiar with the training modalities, including upper-lower limb strength, swimming resistance training, and plyometric training [62] that may be positively transferred to swimming performance [63]. Nevertheless, the lower limbs are only engaged very slightly (12%) in the propulsion of the swimmer [24] and dryland training modalities mostly consist of upper limb exercises that are accompanied by improved swimming performance related to increased upper limb power [63]. Collectively, the lack of ergogenic effect of CAF in jump performance may be attributed to the low specificity of the dryland modalities, which may not have been transferred to the jump performance, although it seems to provide improvement in swimming performance. In fact, previous studies indicated an improvement in jump performance, and such studies enrolled several experimental groups including trained volleyball players [55,56,59], active males with resistance training [48,52], well-trained high jumpers [54], soccer players [57], collegiate athletes [58], and elite Brazilian Jiu-Jitsu athletes [20]. Given the fact that the dryland nature of both team and individual sports comprises speed, repeated sprint ability, agility, balance, muscular strength, acceleration, jumping, and explosive power, the ergogenic effect of the CAF in these studies may have resulted from participants’ responsiveness to the tests identical to the movement patterns and biomechanical demands in their sports. Specifically, success in swimming is mainly dependent upon technical factors [63]; thus, it is not inconsistent to speculate that CAF led to a positive effect only in a test familiar to the experimental group in the present study.

Reaction time (RT) is one of the critical component performance parameters in swimming. In our study, CAF intake appears to have no effect on ART values. Other studies, however, have revealed contradictory results [41,64,65,66,67]. Impey et al. [64,65] reported that CAF ingestion 60 min before exercise improved visual and auditory reaction times. Likewise, a study by Virdinli et al. [68] suggested that CAF mouth rinse increased RT performance in forty-five trained athletes. Accordingly, Church et al. [69] found that acute CAF ingestion had some ergogenicity on visual reaction time in recreationally active adults. A similar result was observed in moderately trained taekwondo athletes, and acute CAF consumption reduced RT [70]. Also, Balko et al. [66] noted that increasing the amount of CAF would enhance reaction time performance by reducing visual and auditory reaction time but not with the lower doses in active sports students. Multiple factors exert an effect on RT, which can be categorized into those associated with the individual and stimuli. Factors contributing to an individual’s response include fatigue, physical state, experience, motivation, sex, and age. Furthermore, another significant factor to consider is the physical attributes of the stimulus, including its duration and level of intensity [71]. Notably, RT performance is a multifaceted phenomenon that should be investigated considering several factors, including cognitive function, physical activity level, and individual responses. Therefore, further studies are required to evaluate its more probable mechanism with different methodological approaches. In highly trained soccer players, 6 mg/kg of CAF intake increased jump performance but did not improve RT [41]. Consistent with our findings, Bottoms et al. [67] indicated that 3 mg/kg of CAF intake did not improve RT in eleven competitive fencers. Since it has been suggested that highly trained athletes may have a higher adaptive response to the ergogenicity of CAF [44], the lack of CAF’s ergogenicity on RT in the present study can be best explained by the training status of the athletes. Although the present study did not reach statistical significance, balance test scores were slightly higher in favor of the CAF condition. Only a few studies have investigated the effect of CAF on postural balance; however, there is evidence that acute CAF ingestion could provide positive effects on static and dynamic balance performance [72,73], while the opposite was suggested in a previous study [74]. These inconsistent results may have resulted from the biomotor abilities of the participants that related to their age, sex, and training status. Accordingly, Kara et al. [73] administered 6 mg/kg CAF in fifteen recreationally active males and examined its effects on balance performance during perceptual-cognitive tasks. Surprisingly, balance scores were lower in CAF conditions compared to water and/or control conditions. It is plausible to think that more consistent results for the balance performance are likely to be obtained via different doses of CAF and experimental designs. Together, these results strongly suggest that further studies investigating the effects of CAF in sports contexts should include various and comprehensive balance tests to provide new insights.

The present study should be evaluated with its limitations: (a) the present study enrolled young (age: 21.3 ± 1.4 years) female swimmers; however, we did not control the menstrual cycle phase, which may have potentially influenced the results; (b) only an acute moderate dose of caffeine was implemented in this study, so it remains unclear whether different doses of caffeine would produce lower or greater performance outcomes; (c) the absence of biochemical analysis in the study prevented us from determining the blood caffeine concentration gathered following CAF intake; and (d) only one lower-body power test was performed in strength evaluation. However, given that the lower limbs only engaged very slightly (12%) in the propulsion of the swimmer [24] and dryland training modalities mostly consist of upper limb exercises, swimming-specific or upper body power tests could have provided more understanding about CAF’s ergogenicity in swimming performance.

## 5. Conclusions

In the present study, we provided findings showing that 6 mg/kg of CAF ingested 60 min before training had no impact on VJ and ART in female swimmers. However, balance test scores were slightly higher in the CAF group compared to the PLA group, and a moderate main effect was observed, although it did not reach significance. Lastly, our novel result indicated that CAF reduces swimming time and improves in-water performance in female swimmers. Together, the present study may indicate that acute CAF intake is effective as an ergogenic aid in short-distance freestyle swimming performance. We recommend that coaches and practitioners incorporate CAF supplementation into swimmers’ nutrition plans before competitions to increase performance.

## Figures and Tables

**Figure 1 nutrients-16-01253-f001:**
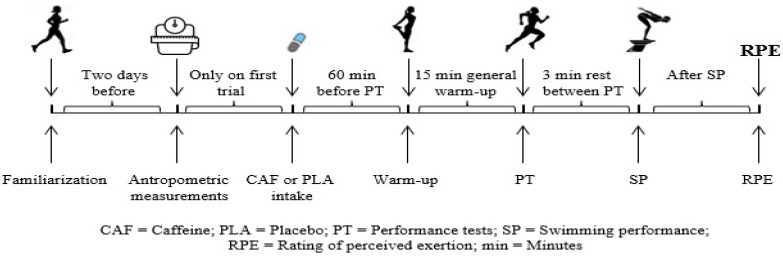
Schematic diagram of the experimental design.

**Figure 2 nutrients-16-01253-f002:**
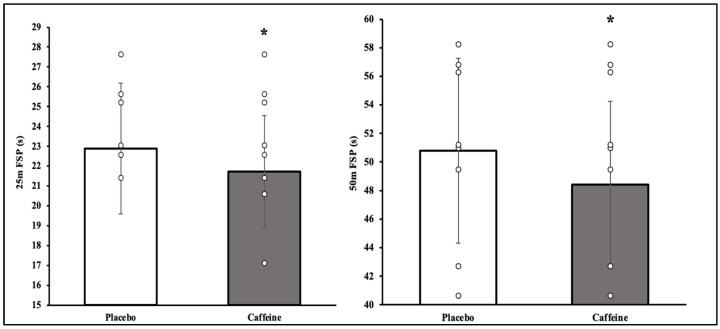
25 m and 50 m freestyle swimming performance. *, significantly different from Placebo and Caffeine.

**Table 1 nutrients-16-01253-t001:** Descriptive information of subjects (*n* = 8).

Variables				X		SD	
Age (yr)				21.3		1.4	
Height (cm)				161.2		7.1	
Body mass (kg)				56.3		6.7	
BMI (kg/m^2^)				21.9		1.3	
Training age (yr)				8.5		1.4	
Habitual consumption of CAF (mg/day)				246.4		111.4	
**Rating of Perceived Exertion (RPE) (A.U.)**							
**PLA (X ± SD)**	**CAF (X ± SD)**	**95% CI**			**d**	**t**	** *p* **
		**LB**	**UB**				
3.0 ± 0.7	3.0 ± 0.9	−0.89	0.89374		0.00	0.000	1.000

X = Mean; SD = Standard deviation; A.U.: Arbitrary units; 95% CI = Confidence Interval; LB = Lower Bound; and UB = Upper Bound.

**Table 2 nutrients-16-01253-t002:** Changes in mean values of CAF and PLA groups.

	Groups						
Variables	PLA	CAF	95% CI		d	t	*p*
	X ± SD	X ± SD	LB	UB			
VJ (cm)	42.6 ± 5.8	41.8 ± 4.1	−2.64	4.14	0.14	0.522	0.618
VJ (WATT)	810.8 ± 92.5	805.3 ± 96.8	−26.39	37.43	0.05	0.409	0.695
Balance (s)	2.2 ± 0.6	1.8 ± 0.4	−0.16	0.81	0.58	1.562	0.162
ART Fastest (ms)	244.2 ± 30.8	217.6 ± 78.6	−41.64	94.89	0.44	0.922	0.387
ART Slowest (ms)	552.0 ± 155.3	544.1 ± 135.5	−184.69	200.44	0.05	0.097	0.926
ART Average (ms)	327.6 ± 50.0	320.3 ± 28.5	−35.93	50.68	0.18	0.403	0.699
ART Deviation (ms)	115.6 ± 55.8	118.2 ± 58.6	−82.39	77.24	0.04	−0.076	0.941
25 m FSP (s)	22.8 ± 3.2	21.7 ± 2.8	0.13	2.18	0.37	2.675	0.032 *
50 m FSP (s)	50.7 ± 6.4	48.4 ± 5.8	0.25	4.47	0.38	2.645	0.033 *

* *p* < 0.05; X = Mean; SD = Standard deviation; d = Cohen’s d effect size; VJ = Vertical jump; FSP = Freestyle swimming performance; 95% CI = Confidence Interval; LB = Lower Bound; and UB = Upper Bound.

## Data Availability

The data presented in this study are available on request from the corresponding author. The data are not publicly available due to privacy restrictions.

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
