# Peer review of "Caffeine Improves Sprint Time in Simulated Freestyle Swimming Competition but Not the Vertical Jump in Female Swimmers"

_nutrients, 2024, doi:10.3390/nu16091253_

Round 1

Reviewer 1 Report

Comments and Suggestions for Authors

Overall, the team of authors has created a fascinating study on the effects of caffeine (known to be an effective ergogenic aid in enhancing sports) on female college-level athletes. Subjects (n=8 per group) took a placebo pill or 6mg/kg caffeine pill and underwent testing of vertical jump (VJ), sprint, balance, agility, and freestyle swimming performance (FSP). The administered acute CAF supplementation improves FSP in 25 and 50 meters, but all other aspects of athletic performance did not achieve statistical significance (p<0.05) in this study. 

The only issues that need clarification from the reviewer's standpoint are the reported p-values and statistical significance in the text and figure legends. In the abstract, it is stated by the authors that CAF supplementation resulted in significantly lower time both in 25m (p= 0.32) and 50m (p= 0.33) of freestyle swimming performance. However, caffeine supplementation resulted in no significant difference in VJ, ART, and RPE (p> 0.05).

It is unclear how p=0.32 and p=0.33 from the abstract (also mentioned on page 7, lines 240-244) are statistically significantly different (or statistically significant). Figures 2 and 3 show the results of freestyle swimming performance, and it is clear that CAF groups are lower in time (s) vs. the placebo group in 25- and 50-meter swimming. Still, the difference is not statistically significant (if the p-values are not below the 0.05 threshold). 

However, later on page 7 (text in lines 251-253), the authors wrote that "No difference was observed in VJ height (42.62 ± 5.80 vs. 41.87 ± 4.18 p=0.001, d= 0.83, table 2) and power (p=0.008, d=0.45, table 2) in CAF and PLA conditions." If the P-values are 0.001 and 0.008, these results must establish that the data has statistical significance, in contrast to what the authors stated. 

On the next page, 8, the results presented in Table 2 show only two p-values below 0.05, and these are the results for 25—and 50-meter freestyle swimming performance (FSP). Table 2 will make the readers (and reviewers) question everything the authors wrote on the previous page and make it very confusing. For example, for the 25m FSP, the p-value listed is =0.032*. However, in the abstract and on page 7, the p-value is listed as 0.32 (is that a typo, and which one is correct, 0.32 or 0.032?). 

The same confusion goes for the VJ heights, which on page 7 in the text in line 251 are listed as 42.62 ± 5.80 vs. 41.87 ± 4.18, p =0.001, d=0.83), however in table 2 on page 8, the VJvalues are different: 42.62 ± 5.80 vs. 41.87 ± 4.18, p =0.618, d=0.14. Please clarify which set of values is correct and update/re-write the text accordingly.

Otherwise, the text is very detailed and describes the methods used well, and the discussion is mostly nicely written and described. The limitations listed at the end of the discussion are very helpful, and overall, this article fits the journal's scope.

Author Response

Dear Reviewer 1,

Time is an extremely precious resource. We, the authors, thank you for the time you gave to study our article and for the advice provided

Reviewer 2 Report

Comments and Suggestions for Authors

The manuscript “Caffeine Improves Sprint Time in Simulated Freestyle Swimming Competition but not the Vertical Jump in Female Swimmers” by Acar is a research article which examined examined the effects of Caffeine (CAF) intake on vertical jump, balance, auditory reaction time (ART), and swimming performance in female swimmers. In a double-blind, cross-over design, eight moderately-trained female swimmers ingested caffeine (CAF) or a placebo (PLA) 60 min before completing vertical jump (VJ), balance, auditory reaction time (ART) and 25/50 m freestyle swimming performance (FSP). The authors found that CAF supplementation resulted in significantly lower time both in 25 m and 50m FSP, but CAF resulted in no significant difference in VJ, ART, and RPE. The authors also found that balance test had almost a non-significant moderate main effect. Therefore, the authors suggested that CAF seems to reduce time in short-distance swimming performances, which could be the determinant of success considering the total time of the race. In general, this review article is critical in this field and contains essential contents. However, I have several comments before this manuscript is accepted for publication.

1. Please explain why this study was conducted by only female.

2. Please add the explanation in the methods section why the concentration of caffeine was 6 mg/Kg in this study.

3. Statistical significance was assessed using t-test. Please add the t values.

Author Response

Dear Reviewer 2,

Time is an extremely precious resource. We, the authors, thank you for the time you gave to study our article and for the advice provided
